# Factors Promoting Clean Energy in Japanese Cities: Nuclear Risks Versus Climate Change Risks

**Ryoko Nakano [1],\*, Tomio Miwa [2],\***  **and Takayuki Morikawa [3]**

[1]  Institute of Global Environmental Strategies, Kanagawa Prefecture 240-0015, Japan
[2]  Institute of Materials and Systems for Sustainability, Nagoya University, Nagoya 464-0819, Japan
[3]  Institute of Innovation for Future Society, Nagoya University, Nagoya 464-0819, Japan; morikawa@nagoya-u.jp
\*  Correspondence: r-nakano@iges.or.jp (R.N.); miwa@nagoya-u.jp (T.M.)

**Abstract:** This article focuses on understanding the factors affecting the subconscious minds of urban citizens in terms of promoting clean energy and deregulation of the electricity sector. Does risk perception related to climate change and nuclear energy effect their choices? Does it differ between cities? A comparative analysis was performed for four cities after the accident at the Tokyo Electric Power Corporation's (TEPCO)'s Fukushima Daiichi Nuclear Power Plant in 2011. This article uses a modeling technique based on surveys gathered in 2012. The results show that nuclear risks had a larger influence than climate-change risks with regards to supporting the deregulation of the electricity sector in TEPCO-serviced cities. Meanwhile, in non TEPCO-serviced cities, nuclear risks were more influential when the proportion of nuclear within the energy mix of the local utility was large. When the proportion was low, climate-change risks had the larger influence. Meanwhile, results from all four cities show that there is indeed a positive causal relationship between citizens' levels of awareness of climate change and energy savings.

**Keywords:** clean energy; cities; climate and nuclear risks; structural equation modeling; Japan

## 1. Introduction

There is growing public support for clean energy policy. However, the reasons underlying this support may vary. This variation is particularly important to policymakers looking to advance and promote new strategies. In the case of Japan, the quest for clean energy became mainstream after the government established multiple regulations and schemes as a result of the accident at the Tokyo Electric Power Corporation (TEPCO)'s Fukushima Daiichi Nuclear Power Plant in the Fukushima prefecture in 2011. The expansion of the feed-in-tariff system (FIT) in July 2012, and the deregulation of the retail electricity market for households and small medium enterprises in July 2016, resulted in there being 805 utilities and 614 retailers as of October 2018 [1]; this was a transition from a market oligopoly, giving consumers the option to choose suppliers/producers who provide sustainable energy in the form of renewables. Energy is central to socioeconomic well-being as well as being the dominant contributor to climate change, which is a risk that needs to be circumvented for the sake of future generations [2]. Climate change risks cannot be realized by the government alone but require a wide range of stakeholders. In this article, we assess one of those stakeholders, citizens, whose acknowledgement of and support for Japan's deregulation of its energy market can help shape this transition. In doing so, we specifically focus on their acknowledgement of climate change risks and assess the 2011 Fukushima nuclear accident's socioeconomic impact. A comparison between citizens in four cities with different geographical, social, and energy characteristics is made.

The remainder of the article is divided into four sections. The next section reviews the relevant literature and discusses our hypotheses. The third section presents an overview of the situation. The fourth section introduces the methods used to test the hypotheses and the findings, and the final section concludes the article.

## 2. Literature Review

Energy futures vary from country to country since they are shaped by multiple factors, including the country's history of nuclear power, resource stock, infrastructure, climate change vulnerability, and the trust between the state and citizens [3]. Germany, for example, has managed to retain 24% of its energy mix, which includes wind, solar, and biomass; this being the result of a movement that started when "antinuclear" activism developed after the Chernobyl accident in 1986. Sweden, another similar case, has gone through several rounds of changes in its energy mix; first, it chose hydropower, then thermal power, whilst returning repeatedly to nuclear, and finally combining this with biomass for district heating [4]. Japan seeks clean energy for two reasons: As a source to enhance "energy security" and reduce vulnerability related to the geopolitics of oil-producing countries, as well as to steadily remove dependence on nuclear as the trust in nuclear regulatory institutions dwindled after Fukushima [3,5]. From the context of a developing country, China's investment in wind and solar is primarily to counter "air pollution" from coal-powered utilities, while India deploys a nationwide solar program to increase availability and affordability in rural regions where demand for energy access needs to be met and energy poverty reduction prevails [6].

On the other hand, different cities' choices of clean energy are site specific, usually trying to address priorities specific to the urban challenges of the city [7,8]. These priorities can be economic, for example, stemming from hopes of pulling out of a recession by introducing a new, clean-energy industry that would increase the economic competitiveness of the city by creating new jobs and fostering regional wealth [9,10]. Others cite health benefits resulting from the reduction in air pollution as being an incentive [11]. Addressing climate risks is, of course, another key factor, which is made appealing when linked to financial subsidies from support programs offered by national governments [10,12,13]. Finally, cities choose to promote clean energy within the context of developing a sustainable community [14] and can be remarkably independent of the national government, as we see in the U.S., where in the 2000s, the federal government's indifference to the Kyoto Protocol prompted forward-looking cities to sign up to meet the U.S.'s environmental target under the Kyoto protocol.

Extensive research has been conducted in a field termed "risk research". Risk perception has been used to describe attitudes and intuitive judgement: It shapes a policymaker's choice for an instrument to promote social transformation; business decisions to restructure unsustainable business practices or a decision to adopt a specific technology; and a consumer's willingness to pay for a service or technology that would change their lifestyles and behavior. Trust for the regulatory institution and socioeconomic benefits increase willingness to accept a new technology [5,15], while environmental harm and perceived costs to counter those harms reduce it [16,17]. Meanwhile, cultural cognition theory holds that subjective risk perceptions are beliefs unconsciously chosen by an individual to support a way of life. Important risks have been identified as dread risk with a high score in the perceived lack of control, dread, and catastrophic potential, as well as those that result from hazards unobservable, unknown, and new [18]. Other studies show risks tend to differ between age, gender, income, education, and basic beliefs [19,20].

This leads us to the central research questions of this article. What lies within the subconscious minds of urban citizens in terms of promoting renewable energy and deregulation of the electricity sector? How does the risk perception related to climate change and nuclear energy affect their choices? More specifically, does it differ between cities and their proximity to a potential risk? Our hypotheses are as follows:

1.  Awareness of nuclear risks should be stronger than that of climate-change risks for areas in which TEPCO offers services;
2.  Awareness of nuclear risks could also be stronger than that of climate-change risks for areas in which the regional utilities use a large percentage of nuclear in the energy mix.

The following sections explore these hypotheses. We first start by defining the two risks that are central to this article: climate change and nuclear energy.

## 3. Energy-Related Risks

### 3.1. Climate Change

Climate change poses a grave and enduring threat to the health and well-being of current and future generations [21]. A risk assessment made by the Ministry of Environment of Japan shows that the temperature levels could rise between 2.1 and 4.0 °C, in which the northern part of Japan would experience a higher rise of temperature, and the southern islands of Kyushu, West Japan, and East Japan could experience a larger number of hot summer days and a higher frequency of strong tropical downpours [22]. This would lead to water shortages in northern Japan. It also indicates a greater frequency of floods, which would lead to heavier causalities from landslides as surface soil is flushed away and the erosion of the ground becomes apparent. The ecosystem would change as animals and vegetation would head north in pursuit of an environment that supports them, while rice production, the staple food for Japanese consumers, would rise in quantity but be reduced in quality.

Recognition of such risks has helped shape Japanese climate and energy policies in the wake of the ratification of the Kyoto Protocol in 2005. A Global Warming Prevention Headquarters was established to implement the Kyoto Protocol, and institutions were set up to support the development of greenhouse gas (GHG) inventories. Energy conservation measures were seen as a key factor in achieving the commitment Japan had set for the Kyoto Protocol; however, having only an array of voluntary targets resulted in a lack of enthusiasm and incentives for domestic industries, who had at the time already taken effective measures as a result of the oil shock of the 1960s. Project offset mechanisms, such as the clean development mechanism (CDM), which allows developed countries to purchase emission reduction credits from abroad, were one of the crucial mechanisms Japan used to keep its commitments.

Japan has currently set a post-2020 $CO_2$ emission target in its Nationally Determined Contributions (NDC), which was incorporated by the cabinet into the Plan for Global Warming Countermeasures in 2015. The target aims to reduce GHG emissions by 26% of 2005 levels by 2030. As a consequence, Japan pledged that 22–24% of the energy mix would need to be renewable energy resources; yet, only 10% was accounted for by renewables as of 2012, and if hydropower is excluded from this, the figure drops to 6% [23].

### 3.2. Nuclear Risks

Japan is known to have used nuclear power as a powerful resource as a result of the lack of fossil fuels, to secure a stable domestic energy supply and reduce environmental concerns (until the Fukushima nuclear accident). Nine regional power utilities existed, of which the share of nuclear in the energy mix was in the range 30–54%. The government chose to keep using nuclear and leave thermal plants idle for economic reasons. The 2010 edition of the nation's Strategic Energy Plan stated that the energy mix in 2030 would consist of 11% coal, 2% oil, and 52% nuclear [24].

The government has steered away from this 2010 target after the Fukushima nuclear accident. Concerns for local safety overrode the reluctant acceptance of nuclear technology as a climate change mitigation measure despite its risks [3]. It was questioned whether this seemingly innovative technology had overcome the high risks of radioactive contamination, which had become a contentious debate.

### 3.2.1. Radiation

Radiation of approximately 1 mSv, which poses a high risk of cancer, was found in the air and oceans surrounding Fukushima at a radius of 30 km immediately after the nuclear accident, which resulted in contamination of dairy and meat products from cattle that fed from the contaminated grass pastures. The radiation levels found in tea leaves, mushrooms, and seaweeds were also examined, leading to a no-buy movement of Fukushima products both domestically and internationally. Residents have moved out of these districts, and farmers and fishermen continue to suffer economically as their products are stigmatized in the market.

As of 2017, an overview of the radiation levels within a 50 km radius of the site, as represented by Iwaki city, are higher than those of Kawasaki and Yokohama that are located approximately 300 km away, as well as Nagoya that is 600 km away, indicating the risk potential of being in close proximity to a nuclear reactor (Figure 1a,b).

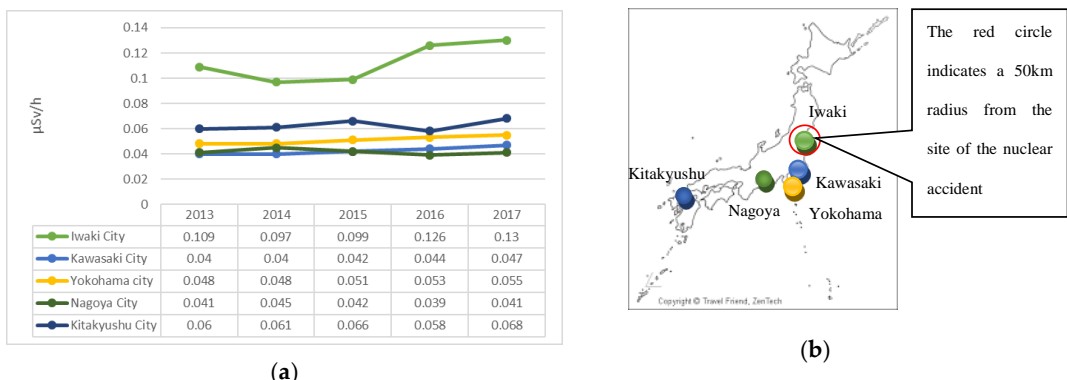

**Figure 1.** (**a**) Radiation level for five cities (2013–2017). Created by the author from data retrieved from the Nuclear Regulation Authority database [25]. (**b**) Location of the five cities.

### 3.2.2. Stress from Evacuation

In response to the radiation, the government ordered the evacuation of areas close to the nuclear accident. The number of evacuees at its peak reached 164,864 in May 2012 and dropped to 57,638 as of July 2017 [26], which is approximately 3% of the total population of the Fukushima prefecture. (Table 1).

**Table 1.** Breakdown of evacuees in May 2012.

| Government Category | Description | Number of Evacuees |
|---|---|---|
| Difficult to return | Residents within 20 km of the nuclear site | 78,000 |
| Restricted residency | Areas with high contamination | 10,000 |
| Preparation required for emergency evacuation | 20–30 km from nuclear site | 58,500 |
| Voluntary evacuation | Residents within 20 km of the nuclear site | 18,000 |

Source: Japan Reconstruction Agency 2017 [27].

In July 2017, a total of 7.34 trillion yen was agreed between TEPCO and the affected individuals and businesses as compensation money for the Fukushima accident with more to come. For an average household (two persons) who used to reside in the region where the government will not allow them to return ("difficulty to return" in Figure 1), the compensation was an average of 36 million yen. Meanwhile, the loss of their house and/or land, plus the additional costs incurred from the purchase of a new home would add up to a total of 74 million yen. It is typical for an evacuee to be under substantial financial stress. This, as well as personal involvement in the evacuation, management,

and clean-up of the aftermath, emerged as the biggest factors for ill health, and not the exposure to radiation [27].

### 3.2.3. Electricity Tariff Hike

During 2012–2016, all eighteen nuclear plants were stopped because of the need to reinstall preventative measures against tsunamis and earthquakes. Once the National Regulation Authority reviewed the regulatory requirements for fresh installments and the operation of existing nuclear power plants, the plants were granted approval. As of December 2017, seven plants have restarted operation or have received approval, and ten are being evaluated of their conformity to the new requirements. One plant (TEPCO's Fukushima Daini Nuclear Plant) has not filed for approval (Figure 2a, Federation of Power Companies of Japan, 2017).

Previously, TEPCO operated nineteen nuclear reactors, and in terms of sheer number, it was the largest among the nine regional utilities, accounting for 31.8% of its energy mix. Kyushu Electric operated six reactors (46.4% of the energy mix) and Chubu Electric managed five (15% of the energy mix).

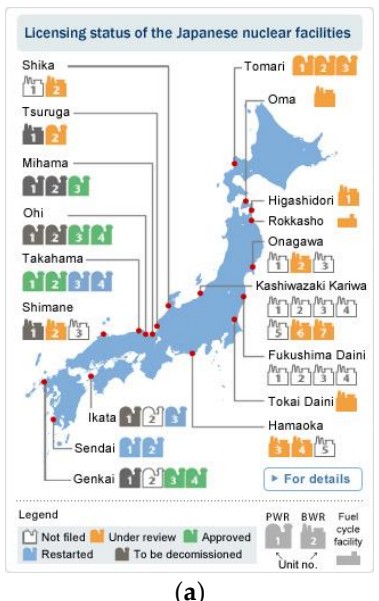

(**a**)

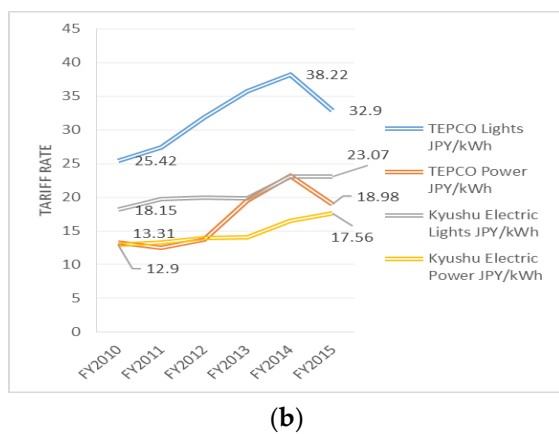

(**b**)

**Figure 2.** (**a**) Map of nuclear power plants (sourced from the Federation of Electric Power Companies of Japan (2017) [28]); (**b**) electricity tariff rates (FY2010–2015) (created by the author from TEPCO and Kyushu Electric financial statements).

TEPCO's tariff rates for 2015 rose by 29.4% and 42.6% from the 2010 rates for lighting and electric power, respectively [29]. Meanwhile, Kyushu Electric's tariff rate also rose by 27.3% and 36.09%, respectively [30,31]. The TEPCO price rise exceeds that of Kyushu Electric, even though the nuclear percentage is higher for Kyushu Electric, hinting at other costs being the cause.

## 4. Research Framework

To test our hypotheses, a study, which compared cities, was conducted using the following two steps:

Step 1): A scoping survey was conducted that included 4000 households in Japan in October 2012 after the nuclear accident. The survey was administered through web-based surveys between October 2012 and November 2012 to 4000 respondents. Several method biases are known to affect a respondent's answers to a survey [32]. Respondents were assured of their anonymity to reduce the possibility for answers to reflect their actual behaviors rather than the politically correct impression they would like to give. The questions were written up with much detail, reducing space for any ambiguity. They

would also appear one-by-one on a screen making the response intuitive and freeing the influence from previous questions. The survey targeted four cities, with the quantity of respondents being selected in proportion to the cities' population (described in Table 2) and their villages. The cities targeted for this study were Yokohama and Kawasaki both located in east Japan, which receive electricity generated by Tokyo Electric (otherwise known as TEPCO), and the cities of Nagoya and Kitakyushu located in west Japan, which are supplied power by other utilities than Tokyo Electric. The four cities represent three of the four major economic districts in Japan; Tokyo (Yokohama and Kawasaki cities), the Aichi prefecture (Nagoya city), and the Fukuoka prefecture (Kitakyushu city). Furthermore, all four cities are known for their citizens' strong environmental actions against water and air pollution, and/or waste management issues in the past. The citizens in the selected regions are, in general, well known for their awareness on environmental policies, and were selected because their behavioral factors lead and represent the country.

Step 2): The data were divided by city. A comparative analysis was conducted between the four cities on the correlation between four factors: perception for nuclear risks, climate change, electricity deregulation, and energy savings. Several questions were used to define each of the four factors, and a confirmatory factor analysis was used on the data to check the correlation between the questions and the groupings.

*4.1. The Data*

As the survey intended to analyze differences across cities, 1000 respondents were taken from four megacities with populations of over one million, namely:

1.　Yokohama—a sprawling suburban commercial port not far from Tokyo, a TEPCO service area;
2.　Kawasaki—an industrial center located on the outskirts of Tokyo, a TEPCO service area;
3.　Nagoya—the third largest metropolitan city situated to the west of Tokyo, a Chubu Electric service area;
4.　Kitakyushu—an industrial city located on Japan's southernmost island, a Kyushu Electric service area.

The respondents were picked randomly from a larger preregistered respondent pool. Those belonging to this pool were offered shopping points with a monetary value of approximately 20 yen in exchange for responding to the survey. Sub-national or local surveys typically sample 500–1000 if they use sub-groups in their analysis [33]. We use the sample size of 1000 per city because the standard deviation is ±3.0% regardless of the population size and is sufficient for this study.

4.1.1. An Overview of the Data

Respondents were between 20 and 69 years old. The mean age was 45.64 years. An effort was made to retain a balance across age groups; the sample was nonetheless skewed towards participants in their 40s, with fewer respondents in their twenties and sixties in all the four cities despite the distribution in the respective location. This may be due to middle-aged respondents having a better availability and technical resources in terms of replying to web surveys.

Another characteristic of the respondents that is thought to be attributable to the design of the survey is the balance of gender, which was slightly tilted towards more men than women, while statistically the percentage of women are known to be somewhat larger.

Household income was divided into five categories (see Table 2). The age of the respondents who are below the 3-million-yen range is slightly higher than the national averages [34]. This may be attributable to the decision to offer monetary compensation for responses, i.e., less wealthy people may be more inclined to answer when there is payment. The slightly higher concentration of lower-income respondents notwithstanding, the distribution of income levels tends to be roughly parallel to national household incomes. That being said, data was not available at the city level and this article must conform to the data at the national level.

Education levels were based on whether respondents indicated that they had formal education, from junior high school to an undergraduate degree and above. According to Japan's national statistics bureau, 51% of students move on to obtain undergraduate degrees [35]. This was indeed very close to the breakdown in the sample, as shown in Table 2. Education-level data at the city level included existing and graduated students and was not clear of their age at the point of the survey. Hence it was difficult to capture respondents who were within the range of this study from city level data.

The questions used to capture social variables included whether the respondents owned or rented a house and the structure of the house. As detached houses are generally more expensive in Japan, the social status of owning one is generally higher than those living in an apartment with a similar number of rooms. Since the respondents enjoyed higher income levels, there was a higher tendency for them owning homes. This did not represent the general distribution, which shows a higher percentage of rented homes in all of the four cities. Finally, with the exception of the city of Kitakyushu, the variable for occupation was distributed evenly in the remaining three cities showing there was no bias there. The sample in Kitakyushu, however, showed a larger number of respondents who were permanently employed. Both the reasons for the bias in the house ownership and the occupation for Kitakyushu might be due to way the survey was conducted: through web-based surveys.

The survey contained a description of the environmental benefits of fossil fuel and renewable energy sources, including nuclear as one option. The challenges for their actual deployment was also clearly mentioned to allow for a balanced and non-biased response.

**Table 2.** Demographic distribution of samples for 2012.

| | | | **Samples** | **%** |
|---|---|---|---|---|
| Gender | Men | | 2258 | 56.5% |
| | Female | | 1742 | 43.5% |
| Age | Average | | 45.6 | |
| Residence | East | Kawasaki | 1000 (1,448,196) | 25.0% |
| | | Yokohama | 1000 (3,703,998) | 25.0% |
| | West | Nagoya | 1000 (2,263,894) | 25.0% |
| | | Kitakyushu | 1000 (974,287) | 25.0% |
| Final education | High school | | 2011 | 50.3% |
| | Undergraduate and above | | 1989 | 49.7% |
| Income | −3 million | | 794 | 19.2% |
| | 3 million–5 million | | 1088 | 27.2% |
| | 5 million–7 million | | 928 | 23.2% |
| | 7 million–10 million | | 706 | 17.1% |
| | 10 million– | | 484 | 12.1% |
| Occupation | Unemployed or temporary | | 1482 | 37.1% |
| | Permanent occupation | | 2518 | 63.0% |
| House ownership | Rented home | | 1376 | 34.4% |
| | Owned home | | 2624 | 65.6% |

Note: ( ) is the total population according to surveys conducted at each municipality between 2012–2013.

4.1.2. A Comparison between Cities

A comparison between the four cities shows that interests in climate change are consistent throughout, in which over 60% were either a little or very interested (Table 3). Engaging the public regarding climate change issues is a particularly challenging endeavor because the impacts are often perceived to be uncertain, in the distant future, and not personally relevant [36]. Hence, the awareness level shown indicates that the strong communication efforts by the Ministry of Environment have paid off. It is also notable that one fourth of citizens for all four cities were not quite sure on whether to halt nuclear or not, while 22–28% thought that it should be stopped immediately.

A chi-square test (χ2 test) was conducted to observe possible differences in the distribution of responses between the cities serviced by TEPCO (i.e., Kawasaki and Yokohama) versus the others (Nagoya and Kitakyushu). The two groups of citizens differ in the following ways: Their energy-saving behavior after Fukushima; their habit of checking their energy bills; the level of support for deregulation of the electricity market; and the level of support for decentralization of the energy system (Table 3).

More energy-saving behavior was prompted by the Fukushima accident in both Kawasaki (76%) and Yokohama (67%), while respondents in Nagoya (73%) and Kitakyushu (75%) stated that it did nothing for them in terms of changing their behavior. This trend continued into the following year although the number of energy-saving citizens in Nagoya and Kitakyushu did rise. The habit of checking electricity bills is common in all four cities, while there does not seem to be much focus on actually understanding which appliances in their homes consume more electricity and should be targeted for efficient usage.

The behavior of normally checking energy bills also differed between the two groups in 2012: The tendency to do so was stronger in Kawasaki (71.4%) and Yokohama (72.4%), compared to their counterparts in Nagoya (68.0%) and Kitakyushu (66.5%). The variance could no longer be found in 2013 in which the percentage of citizens in Kawasaki and Yokohama doing so dropped.

A promising view was uncovered from the survey results, as all four cities show considerable interest in climate change and are supportive of halting nuclear power either immediately or by 2030 regardless of the grouping and this trend continues at least for the two years of this survey: 2012 and 2013. There was variance in their favor for electricity market deregulation, which is considerably stronger in the TEPCO group (Kawasaki (72.8%) and Yokohama (67.9%)) compared to Nagoya (58.2%) and Kitakyushu (56.3%) and this continued into 2013, the following year.

The results, however, pertain to the four megacities in Japan and do not represent the Japanese citizens in general. Furthermore, the comparison here is to compare the TEPCO group with the non-TEPCO group with the four cities, although they might not represent the citizens of the other cities within the same groups.

**Table 3.** Reply per city.

| Question | χ2 | Kawasaki | | Yokohama | TEPCO | Nagoya | Kitakyushu | Non-TEPCO |
|---|---|---|---|---|---|---|---|---|
| Do you have interest in climate change issues? | | No | 4.1% | 3.5% | 3.8% | 3.9% | 3.6% | 3.7% |
| | | Not much | 7.6% | 5.1% | 6.3% | 8.1% | 7.5% | 7.8% |
| | | I am indifferent | 19.2% | 18.5% | 18.9% | 24.3% | 24.4% | 24.4% |
| | | Yes, a little | **51.8%** | 50.6% | **51.2%** | 48.0% | 47.6% | **47.9%** |
| | | Yes, definitely | **17.3%** | 22.1% | **19.7%** | 15.5% | 16.7% | **16.2%** |
| What are your thoughts on generating energy from nuclear utilities? (Should they stop? When?) | | Use beyond 2050 | 10.6% | 10.6% | 10.6% | 13.2% | **16.1%** | **14.7%** |
| | | Stop by 2050 | 21.8% | 21.0% | 21.4% | 8.9% | 7.3% | 8.1% |
| | | Stop by 2040 | 10.2% | 9.9% | 10.0% | 7.0% | 7.1% | 7.1% |
| | | Stop by 2030 | **6.3%** | **6.4%** | **6.4%** | 25.5% | 25.9% | 25.7% |
| | | Stop immediately | **28.5%** | **28.8%** | **28.6%** | 23.0% | 24.2% | 23.6% |
| | | I am not sure | 22.5% | 23.3% | 22.9% | 22.5% | 19.4% | 20.9% |
| Did the Fukushima nuclear accident prompt you to save energy in your home, office, and community? | * | Definitely not | 3.3% | 5.2% | 3.3% | **19.8%** | **21.8%** | **20.9%** |
| | | Not much | 5.2% | 6.8% | 5.0% | **18.5%** | **18.8%** | **18.7%** |
| | | I am indifferent | 14.4% | 19.9% | 13.9% | **36.8%** | **36.0%** | **36.5%** |
| | | Yes a little | **40.4%** | **41.9%** | 39.8% | 17.8% | 17.6% | 17.7% |
| | | Yes definitely | **36.7%** | **26.2%** | 38.0% | 6.9% | 5.7% | 6.3% |
| Do you normally check your monthly electricity bills? | * | No | 28.6% | 27.5% | 28.1% | 31.8% | 33.4% | 32.7% |
| | | Yes | 71.4% | 72.4% | 71.9% | 68.0% | 66.5% | 67.3% |
| Are you aware of the electricity consumption for each appliance? | | No | 64.3% | 63.9% | 64.1% | 66.3% | 67.8% | 67.1% |
| | | Yes | 35.7% | 36.1% | 35.9% | 33.6% | 32.1% | 32.9% |
| Do you support the feed-in-tariff system (FIT)? | | Definitely not | 10.4% | 11.5% | 10.9% | 8.4% | 10.2% | 9.3% |
| | | Not much | 13.3% | 12.4% | 12.8% | 15.0% | 14.8% | 14.9% |
| | | I am indifferent | 45.7% | 44.5% | 45.1% | 47.7% | 46.6% | 47.2% |
| | | Yes a little | 24.2% | 23.5% | 23.9% | 20.9% | 22..4 % | 21.7% |
| | | Yes definitely | 6.5% | 8.0% | 7.2% | 7.8% | 5.9% | 6.8% |

**Table 3.** *Cont.*

| Question | χ2 | Kawohama | | Yokohama | TEPCO | Nagoya | Kitakyushu | Non-TEPCO |
|---|---|---|---|---|---|---|---|---|
| Do you welcome the deregulation of the retail electricity market? | | Not necessary | 1.1% | 1.8% | 1.4% | 2.9% | 4.5% | 3.2% |
| | * | Not really needed | 4.0% | 5.4% | 4.7% | 6.3% | 7.3% | 6.8% |
| | | I am indifferent | 22.0% | 24.9% | 23.5% | 32.6% | 32.8% | 32.7% |
| | | Yes probably | **38.1%** | **35.8%** | **37.0%** | **33.1%** | **32.4%** | **32.8%** |
| | | Yes definitely | **34.7%** | **32.1%** | **33.5%** | **25.1%** | **23.9%** | **24.5%** |
| Do you think a decentralized energy system should be promoted? | | No | 1.5% | 2.8% | 2.1% | 2.3% | 2.5% | 2.4% |
| | * | Not really needed | 3.9% | 4.3% | 4.1% | 4.9% | 5.0% | 4.9% |
| | | I am indifferent | 9.7% | 28.6% | 29.2% | 34.7% | 33.3% | 34.0% |
| | | Yes, probably | **47.5%** | **47.5%** | **47.5%** | **44.6%** | **44.3%** | **44.5%** |
| | | Yes definitely | **17.4%** | **16.8%** | **17.1%** | **13.4%** | **14.8%** | **14.2%** |

Note: The χ2 distribution was compared for Tokyo Electric Power Corporation (TEPCO) versus non TEPCO-serviced areas. The asterisks (*) indicate that the χ~2 is statistically different from zero at the 5% significance level.

### 4.2. Structural Equation Modeling

Over 60% in all four cities favor decentralized energy systems. However, again the level of support was stronger for the TEPCO group (Kawasaki (64.9%) and Yokohama (64.3%)), followed by Nagoya (58.0%) and Kitakyushu (59.1%).

This article uses structural equation modeling (SEM) to analyze the factors affecting the inclination to support electricity deregulation. Here, the modeling results also show a difference between TEPCO-serviced cities versus non TEPCO-serviced cities.

The subjective views of the respondents as defined through the surveys were translated into their subconscious attitudes. We defined the factors that determine citizen's choices and behaviors as being respondents' "level of awareness of climate change", "level of awareness of nuclear risks", "support for the deregulation of the electricity market", and "level of awareness of energy savings", and categorized them into latent variables under the SEM. SEM is a methodology for representing, estimating, and testing a relationship between variables (measured variables and latent constructs). The model combines confirmatory factor analysis (CFA) and path model analysis; techniques that are common to factor analysis and multivariate regression to aggregate and group observed variables of similar characteristics into constructs/latent variables, and then see how the different constructs/latent variables relate to each other [31,37,38]. The latent variables are explained through a combination of two different models: 1) The measurement of an individual's views and responses to a question under the measurement model; and 2) the individual's social and economic characteristics under the structural model. The following formula was used:

$$\text{(Measurement model) } \mathbf{y}_i = \mathbf{K}\boldsymbol{\eta}_i + \boldsymbol{\Lambda}\mathbf{x}_i + \varepsilon_i, \tag{1}$$

$$\text{(Structural model) } \boldsymbol{\eta}_i = \mathbf{B}\boldsymbol{\eta}_i + \boldsymbol{\Gamma}\mathbf{x}_i + \zeta_i, \tag{2}$$

where, for each individual $i$, $\boldsymbol{\eta}_i$ is the latent variable; $\mathbf{x}_i$ is an objective variable; $\mathbf{y}_i$ is the observed variable that indicates subjective views; $\mathbf{B}$, $\boldsymbol{\Gamma}$, $\mathbf{K}$, and $\boldsymbol{\Lambda}$ stand for unknown parameters; and $\zeta_i$, $\varepsilon_i$ represent residuals. The coefficients between the subjective views and the latent variables can be found in Table 4. As shown in this table, all of the path coefficients were statistically different from zero.

**Table 4.** Measurement model.

| Latent Variables | Subjective Views Observed Variables | Kawasaki | Yokohama | Nagoya | Kitakyushu |
|---|---|---|---|---|---|
| Level of awareness of climate change | Do you have an interest in climate change issues? | 0.39 * | 2.08 * | 0.35 * | 1.40 * |
| Level of awareness of nuclear related risk | What are your thoughts on generating energy from nuclear utilities? (Should they stop, when?) | 1.42 * | 1.77 * | 2.67 * | 4.97 * |
| | Did the Fukushima nuclear accident prompt you to save energy at your home, office, and community? | 1.00 * | 1.00 * | 1.00 * | 1.00* |
| Support for deregulation of electricity market | Do you support the FIT? | 1.00 * | 1.00 * | 1.00 * | 1.00 * |
| | Do you welcome the deregulation of the retail electricity market? | 1.47 * | 1.44 * | 1.07 * | 1.77 * |
| | Do you think a decentralized energy system should be promoted? | 2.03 * | 1.66 * | 1.39 * | 1.78 * |
| Level of awareness of energy savings | Do you normally check your monthly electricity bills? | 1.00 | 1.00 | 1.00 | 1.00 |
| | Are you aware of the electricity consumption for each appliance? | 0.42 * | 1.15 * | 0.25* | 0.40 * |

Note: The asterisks (*) indicate that the coefficients are statistically different from zero at the 5% significance level.

Table 5 is the structural model, which shows that gender has an impact on energy-related decisions. Females in all four cities were more conscious and welcomed an early removal of nuclear power from their energy mix. Meanwhile, support for electricity deregulation was strong for males in all cities with the exception of Nagoya, in which gender did not have a significant impact. As stated in Section 4.1 there is a larger tendency for males in the sample for all cities, and this might have had a slight impact on the outcome to this last variable. Higher education levels also seemed to be factor with the exception of Kawasaki city, although the question arose of why people with lower levels of education would be more supportive in Nagoya.

**Table 5.** Structural model.

| Latent Variable | Objective Variable | Kawasaki | Yokohama | Nagoya | Kitakyushu |
|---|---|---|---|---|---|
| Level of awareness of climate change | Gender | 0.6 * | - | - | 0.1 * |
| | Age | **0.03 *** | **-** | **0.05 *** | **0.01 *** |
| | Income | 0.1* | - | 0.09 * | - |
| | Education | 0.2* | - | 0.2 * | - |
| | Rooms | - | - | 0.1 * | - |
| | House age | −0.1* | - | −0.2 * | - |
| | House owner | - | - | 0.6 * | - |
| | House structure | 0.03 * | - | −0.03 * | - |
| | Occupation | - | - | - | - |

**Table 5.** *Cont.*

| Latent Variable | Objective Variable | Kawasaki | Yokohama | Nagoya | Kitakyushu |
|---|---|---|---|---|---|
| Level of awareness of nuclear related risk | Gender | **0.3** * | **0.2** * | **0.4** * | **0.1** * |
| | Age | 0.006 * | 0.01 * | - | - |
| | Income | - | - | −0.1 * | - |
| | Education | - | - | −0.1 * | - |
| | Rooms | - | - | - | - |
| | House age | - | - | - | - |
| | House owner | - | - | - | - |
| | House structure | - | - | - | - |
| | Occupation | - | - | 0.1 * | - |
| Support for deregulation of electricity market | Gender | **−0.2** * | **−0.1** * | **-** | **−0.08** * |
| | Age | −0.01 * | - | −0.03 * | - |
| | Income | - | - | −0.07 * | - |
| | Education | - | **0.06** * | −0.07 * | **0.03** * |
| | Rooms | - | - | −0.09 * | - |
| | House age | 0.05 * | - | 0.1 * | - |
| | House owner | - | - | −0.3 * | - |
| | House structure | - | - | - | - |
| | Occupation | - | - | - | - |
| Level of awareness of energy savings | Gender | - | - | - | - |
| | Income | - | - | - | - |
| | Education | - | −0.03 * | - | - |
| | Rooms | −0.01* | - | - | - |
| | House age | – | - | - | - |
| | House owner | - | - | - | - |
| | House structure | – | – | 0.02 * | - |
| | Occupation | – | – | - | - |

Note: The asterisks (*) indicate that the coefficients are statistically different from zero at the 5% significance level. Gender (0: male 1: female), Occupation (0: temporary 1: permanent), Education (0: junior high 1: high 2: junior university 3: undergrad and above).

Figure 3 shows the relationship between the latent variables. We can see the impact climate change awareness levels and nuclear risk awareness levels have on electricity deregulation and energy savings. This paper used two indices for diagnostic testing: RMSEA (root mean square of error of approximation) and GFI (goodness of fit). The RMSEA is the fit statistic reported in the LISREL program and shows us how well the model, with unknown but optimally chosen parameter estimates, would fit the population's covariance matrix [39]. A range between 0 and 0.07 seems to be the general consensus amongst authorities in this area [40,41]. For GFI, an omnibus cut-off point of 0.90 and over is recommended [42].

Regarding Kawasaki and Yokohama, the results indicate that while both nuclear risks and climate change were strong factors influencing respondents to support electricity market deregulation, the former was the stronger of the two. This might be the result of three factors: First, both cities are TEPCO-serviced areas and experienced tariff hikes larger than their counterparts; this could have had a direct impact on the respondents. The reliance on nuclear was strong (31.8%) and the operating costs for thermal plants reactivated as alternatives were costly compared to nuclear (for which most were sunk costs); second, respondents were already seeking alternative energy sources and a wider variety of choices for their energy supply to reduce dependency on a single service supplier; third, because it was not clear how much of the substantial compensation charges brought to TEPCO from evacuees would strike the utility financially. On the other hand, the debate on climate change is widely accepted and a strong factor affecting respondents in both municipalities (Figure 3a,b).

Meanwhile, in the case of Nagoya, climate change was definitely the stronger of the two factors, showing a different outcome from Kawasaki and Yokohama (Figure 3c). This might be due to the distance from the Fukushima Daiichi Nuclear Power Plants, and the fact that Chubu Electric was

the service provider and not TEPCO. The lower reliance on nuclear power (15%) is also a factor that might have reduced the influence of nuclear risks on respondents' support for deregulation. Although speculative, the memory of the Conference of the Parties (COP) 10 for the Convention for Biodiversity held in Nagoya in October 2010 only six months before the nuclear disaster and two years before the distribution of this survey might have had a positive lingering effect on respondents' awareness of global environment conservation (i.e., climate change) and influenced the positive decision for deregulation.

In the case of Kitakyushu, both nuclear risks and climate change were significant factors influencing deregulation, although the strength of the impact was less than that of Kawasaki and Yokohama (Figure 3d). The city is the farthest of the four cities from the nuclear accident and the level of nuclear within the energy mix is larger than the TEPCO-serviced areas (46.4%). Kitakyushu city is widely known for its involvement in climate-change-related technology transfer with other cities in South East Asia, which has encouraged the city to maintain and enhance its image both domestically and internationally. Such efforts came after the city was able to overcome the heavy air pollution that prevailed in the atmosphere due to the use of fossil fuel without adequate treatment. For this reason, the city is not alien to the socioeconomic troubles that occur as a result of environmental harm.

Lastly, a notable finding from these models was the positive causal relationship that level of awareness of climate change had with energy savings in all four cities. Meanwhile, nuclear risks were a negative causal factor influencing energy savings with the exception of Kawasaki, which showed a positive relationship. This might mean that people in general were not intuitively connecting nuclear risks with their own energy consuming behavior, while for climate change, so much effort has been made to educate the population that the connection in the subconscious mind was effortless. An analysis of this last outcome should be performed with more historical information, which might help explain the difference between Kawasaki and the other three cities.

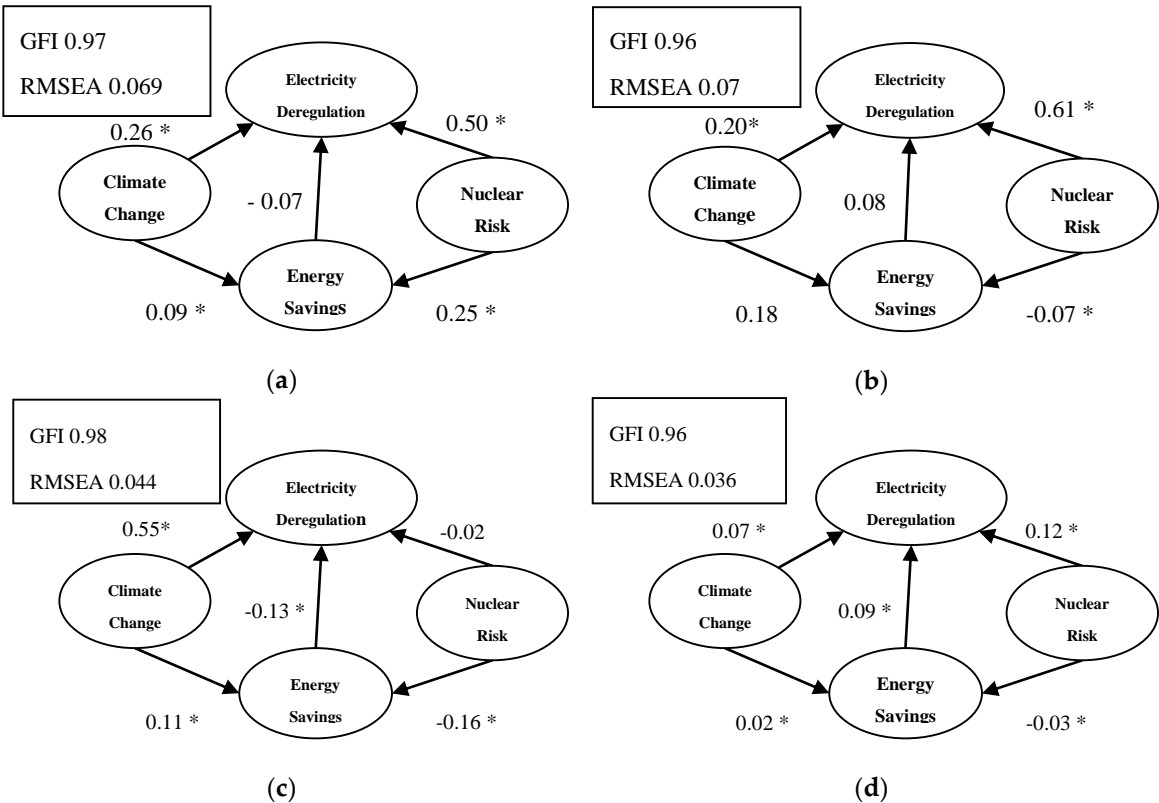

**Figure 3.** Relationship between latent variables: (**a**) Kawasaki; (**b**) Yokohama; (**c**) Nagoya; (**d**) Kitakyushu. The asterisks (*) indicate that the *t* values of the coefficients are statistically different from zero at the 5% significance level.

## 5. Conclusions

The Sustainable Development Goals call for securing access to affordable, reliable, sustainable, and modern energy for all by 2030, with the Paris Agreement for Climate Change also focusing on similar results. In this article, we tried to define what factors would encourage citizens (key actors in this effort) to support a transition to clean energy. We selected two hypotheses and a range of survey questions to evaluate them. The first hypothesis is nuclear risks should have a larger influence than climate change risks for areas in which TEPCO offers services; the second, nuclear risks could also have a larger influence for areas in which the regional utilities use a large percentage of nuclear in their energy mix. The first hypothesis was supported in the TEPCO-serviced cities of Kawasaki and Yokohama. The second hypothesis was supported in Kawasaki, Yokohama, and Kitakyushu. The study also revealed urban citizens residing in TEPCO-serviced areas were more positive compared to their counterparts in non-TEPCO areas in terms of their energy saving behavior after Fukushima; their habit of checking their energy bills; the level of support for deregulation of the electricity market; and the level of support for decentralization of the energy system.

Citizens do show a psychological and behavioral difference between the four cities of Kawasaki, Yokohama, Nagoya, and Kitakyushu that can be explained from a city's distance to a certain risk. The risk perception should be considered and reflected in the messages used to promote local clean energy generation. It should also be a key factor for businesses as sound environmental sustainability gradually becoming a criterion to gain funding and consumer recognition. Furthermore, a general current that runs through all of the cities studied was the relevance of gender. Females in all four cities were more aware and welcomed an early removal of nuclear power from their energy mix. For this reason, gender-sensitive messages should also be considered.

While the results were obtained from the data of the four megacities in Japan and do not represent the Japanese citizens in general, this article contributes to the literature in two ways: it adds to risk perception theory with insights into an independent variable that can be mainstreamed, "physical distance", adding to the discussions in the climate change arena on global versus local risks, offering insights; and it offers analysis to energy transitions in a particular point in time which is at the eve of a policy deregulation that is about to happen.

This analysis of the acceptance and support for energy transitions is still the first step. Future research will focus on how risk perception in each city has actually influenced the energy deregulation in terms of new retail services for electricity for renewable energy deployment and energy efficiency.

**Author Contributions:** Conceptualization, R.N. and T.M. (Tomio Miwa); methodology, R.N. and T.M. (Tomio Miwa); validation, R.N. and T.M. (Tomio Miwa); formal analysis, R.N and T.M. (Tomio Miwa); investigation, R.N.; resources, R.N.; data curation, R.N.; writing—original draft preparation, R.N.; writing—review and editing, T.M. (Tomio Miwa); visualization, R.N.; supervision, T.M. (Tomio Miwa) and T.M (Takayuki Morikawa).

**Funding:** This research was conducted with the generous funding for the project the Environment Research and Technology Development Fund (ERTDF) from the Ministry of Environment Japan (grant number E1105 between the years 2011-2014); Kanagawa Prefecture Grant (between the years 2012-2014).

**Acknowledgments:** Special appreciation goes to Eric Zusman for his insights and continued support that was instrumental to this article.

**Conflicts of Interest:** The authors declare no conflict of interest.

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
