# Peer review of "Factors Promoting Clean Energy in Japanese Cities: Nuclear Risks Versus Climate Change Risks"

_sustainability, doi:10.3390/su11246918_

Round 1
Reviewer 1 Report
Review Report
to the paper entitled
“Factors promoting clean energy in Japanese Cities: nuclear risks versus climate change risks”
Major Comments:
Please substantiate why the given 4 cities are chosen? Are they representative for the region or for the country? How the needed sample size is chosen? Have you used the sample size formula for a certain purpose? Have you conducted the representativeness test to check if your sample is representative for the region/country? Do not you think that survey results (2012) directly after the incident might be biased since the accident just happened and it might have lagged decreasing impact on habitants? Have you conducted the diagnostic tests for your model? Please discuss or note about them. If the model did not pass these tests the results are not interpretable.
Minor Comments:
Line 217. Should not it be table 2? Line 227. Should not it be table 2? Line 247. Should not it be table 3? Please cite the structural equation model (SEM).
Author Response
Dear Reviewer,
Thank you very much for your comments to which I would like to reply one by one.
1) Please substantiate why the given 4 cities are chosen? Are they representative for the region or for the country?
Answer) The four cities represent the 3 of the 4 major economic districts in Japan; Tokyo (Yokohama and Kawasaki cities), Aichi prefecture (Nagoya city) and Fukuoka prefecture (Kitakyushu city). Furthermore, all four cities are known for their citizens’ strong environmental actions against water and air pollution, and/or waste management issues in the past. The citizens in the selected regions are, in general, well known for their awareness on environmental policies, and were selected from the assumption that their behavioral factors are seen to represent the country.
2) How the needed sample size is chosen? Have you used the sample size formula for a certain purpose? Have you conducted the representativeness test to check if your sample is representative for the region/country?
Answer) According to Sudman (1976), sub-national or local surveys typically sample 500-1000 if they use sub-groups in their analysis. We use the sample size of 1000 per city because the standard deviation is +/- 3.0% regardless of the population size of each respective city.
3) Do not you think that survey results (2012) directly after the incident might be biased since the accident just happened and it might have lagged decreasing impact on habitants?
A similar survey conducted in 2013 the following year shows energy saving behavior in Nagoya and Kitakyushu did rise. Furthermore, all four cities show considerable interest in climate change and are supportive of halting nuclear power either immediately or by 2030 regardless of the grouping and this trend continues also for both 2012 and 2013.
Meanwhile, favor for electricity market deregulation which is considerably stronger in the TEPCO group (Kawasaki (72.8%) and Yokohama (67.9%)), compared to Nagoya (58.2%) and Kitakyushu (56.3%) in 2012 also continued into 2013.
4) Have you conducted the diagnostic tests for your model? Please discuss or note about them. If the model did not pass these tests the results are not interpretable.
For the diagnostic testing, this paper used two indices: GFI (goodness of fit) and RMSEA (root mean square of error of approximation). The RMSEA is the fit statistic reported in the LISREL program and shows us how well the model, with unknown but optimally chosen parameter estimates would fit the populations covariance matrix (Byrne, 1998). A range between 0 and 0.07 (Hu and Bentler, 1999; Steiger, 2007) seems to be the general consensus amongst authorities in this area. For GFI, an omnibus cut-off point of 0.90 and over is recommended (Miles and Shevlin, 1998). The last figure shows the GFI and the RMSEA for each city model and illustrates they hold.
Thank you.
Reviewer 2 Report
The paper presents a statistical method to individuate the factors that determine the awareness of people for climate changes on one side and the nuclear risk on the other side. The method well describes how the data are collected and then statistically interpreted. The interviews have been performed in four different cities of Japan, and the data have been collected less than one year after the Fukushima accident. The hypothesis is that one that many factors affect the psychological perception of the risks connected to the use and production of the electrical energy, such as the climate change and the nuclear accident.
Comments:
the sample (1000 interviews) is very low with respect to the sampled population. some validation should be adopted to verify the representativity of the sample, but none is described the set of considered variables is not adequately motivated the limited number of cities (four) limits the reliability of the findings, especially considering the small differences of percentageminor points:
Line 52: wrong name of Chernobyl and wrong date of the accident
Line 67: Others for Other
Line 169: "has not filed"
caption of table 2 duplicated
Author Response
Dear Reviewer,
Thank you very much for your comments. Kindly allow me to respond to them one by one.
1) The sample (1000 interviews) is very low with respect to the sampled population. some validation should be adopted to verify the representativity of the sample, but none is described the set of considered variables is not adequately motivated the limited number of cities (four) limits the reliability of the findings, especially considering the small differences of percentage
-> According to Sudman (1976), sub-national or local surveys typically sample 500-1000 if they use sub-groups in their analysis. We use the sample size of 1000 per city because the standard deviation is +/- 3.0% regardless of the population size of each respective city.
2) Line 52: wrong name of Chernobyl and wrong date of the accident
-> Thank you I will correct the name and the date upon submission of the manuscript.
3) Line 67: Others for Other
-> Thank you I will correct upon submission of the manuscript.
4) Line 169: "has not filed"
-> Thank you I will correct upon submission of the manuscript.
5) caption of table 2 duplicated
-> Thank you I will correct upon submission of the manuscript.
Reviewer 3 Report
Review of the article "Factors promoting clean energy in Japanese Cities: nuclear risks versus climate change risks". Points for correction:1. In the article, the authors consider two hypotheses:
a) Awareness of nuclear risks should be stronger than climate change for areas in which TEPCO offers services.
b) Awareness of nuclear risks could also be stronger than climate change for areas at which the regional utilities use a large percentage of nuclear in the energy mix. In my opinion, the answers to both hypotheses are obvious but only when considering local communities.
The nuclear risk that is the subject of the study (that is the subject of consideration of these two hypotheses), can be considered as local threat that applies especially to highly developed countries.
Climate change is global problem and applies to highly developed countries, low developed countries, and undeveloped countries. Climate change is global problem that also applies to all living forms, not just humans but also animals and plants.
From this point of view climate change is always bigger problem that local nuclear risk. So, taking into account local communities I agree with the Authors. Taking into account life on our Earth I cannot agree with the Authors. 2. Linguistic correction is required, there are a lot of linguistic errors - the article should be corrected by a native speaker.
Author Response
Dear Reviewer,
Thank you very much for your comments. Kindly allow me to respond to each comment as follow
1) In the article, the authors consider two hypotheses:
a) Awareness of nuclear risks should be stronger than climate change for areas in which TEPCO offers services.
b) Awareness of nuclear risks could also be stronger than climate change for areas at which the regional utilities use a large percentage of nuclear in the energy mix. In my opinion, the answers to both hypotheses are obvious but only when considering local communities.
The nuclear risk that is the subject of the study (that is the subject of consideration of these two hypotheses), can be considered as local threat that applies especially to highly developed countries.
Climate change is global problem and applies to highly developed countries, low developed countries, and undeveloped countries. Climate change is global problem that also applies to all living forms, not just humans but also animals and plants.
From this point of view climate change is always bigger problem that local nuclear risk. So, taking into account local communities I agree with the Authors. Taking into account life on our Earth I cannot agree with the Authors.
-> Thank you for your comments. I also support your last point that climate change is a bigger problem taking in the severity of the global impact it is causing. It is now, however, not only a global issue since the weather disruptions caused by climate change are also very much local, hence it has both dimensions.
This article attempts to show how risk (nuclear and climate change) would allow citizens to immediately accept rapid energy deregulation regardless of the additional financial burdens to each individual in a country that relied on nuclear for its base load.
2). Linguistic correction is required, there are a lot of linguistic errors - the article should be corrected by a native speaker.
-> Thank you. I have had the manuscript edited and corrected by a native speaker.
Round 2
Reviewer 1 Report
Review Report (round 2)
to the paper entitled
“Factors promoting clean energy in Japanese Cities: nuclear risks versus climate change risks”
Major Comments:
Have you conducted the representativeness test to check if your sample is representative for the region/country? Goodness of fit and RMSE are not diagnostic tests, they are the quality measures of the model. Please, provide and discuss the diagnostic tests results. Discuss the rationale behind of choosing 1000 as a sample size in the text.
Minor Comments:
Please cite the structural equation model (SEM).
Author Response
Have you conducted the representativeness test to check if your sample is representative for the region/country? Goodness of fit and RMSE are not diagnostic tests, they are the quality measures of the model. Please, provide and discuss the diagnostic tests results. Discuss the rationale behind of choosing 1000 as a sample size in the text.
è Thank you very much for your comments. First, kindly allow us to state the model is looking at the four mega-cities in Japan respectively and do not represent the Japanese citizens in general. Furthermore, the comparison here is to compare the TEPCO group with the non-TEPCO group with the four cities, although they might not represent the citizens of the other cities within the same groups.
è Having said that we did conduct a chi-square test for age and gender at the city level which showed there was a slight selection bias that might be attributable to the survey being conducted by the web. The number of samples for men in their 40s was shown to be larger than the general distribution. This might have had some influence to the stronger support male respondents showed in terms of electricity deregulation. Since the respondents enjoyed higher income levels, the sample showed there was a higher tendency for them owning homes. This did not represent the general distribution with a higher percentage of rented homes in all of the four cities. Finally, with the exception of the city of Kitakyushu, the variable for occupation was distributed evenly in the remaining three cities showing there was no bias there.
è Meanwhile, stratified data at the city level was not available for income distribution, and it was challenging to find data for education levels specific for the range of respondents for this survey.
This has all been added in the revised draft pp7-9 and the conclusion pp15.
Reviewer 2 Report
The authors improved the writing of the paper, and most part of typos have been fixed.
Concerning the method, some perplexities still remain, for instance the fact that in the interviews to the samples the nuclear energy is presented as th unique suitable solution to prevent the emissione of GHGs, which could polarize the people toward a predefined answer. On the other hand, the paper well describes a method for the sociological analysis on sensitive topics such as energy and climate change
Author Response
Concerning the method, some perplexities still remain, for instance the fact that in the interviews to the samples the nuclear energy is presented as th unique suitable solution to prevent the emissione of GHGs, which could polarize the people toward a predefined answer. On the other hand, the paper well describes a method for the sociological analysis on sensitive topics such as energy and climate change
-> Thank you for your comments. The survey contained description of the environmental benefits of fossil fuel and renewable energy sources including nuclear as one option. The challenges for their actual deployment was also clearly mentioned to allow for a balanced and non-biased response.
This has been added in the revised draft pp8
Reviewer 3 Report
Thank you for these corrections. The authors addressed correctly to all my comments and concerns. Now the article is much better and I think that it can be published in Sustainability.Author Response
Thank you for these corrections. The authors addressed correctly to all my comments and concerns. Now the article is much better and I think that it can be published in Sustainability.
-> Thank you very much for your comments.
Round 3
Reviewer 1 Report
The authors have addressed my comments.